# Absenteeism at Two Occupational Health Services in Belgium from 2014 to 2021

**DOI:** 10.3390/ijerph20043660

**Published:** 2023-02-18

**Authors:** Ilse Moerland, Nouchka Vervaet, Lode Godderis, Mathieu Versée, Marc Du Bois

**Affiliations:** 1IDEWE, External Service for Prevention and Protection at Work, Interleuvenlaan 58, 3001 Heverlee, Belgium; 2Cohezio, External Service for Prevention and Protection at Work, Bisschofsheimlaan 1-8, 1000 Brussels, Belgium; 3Centre for Environment and Health, Department of Public Health and Primary Care, KU Leuven, 3000 Leuven, Belgium; 4Mental Health and Wellbeing Research Group, Department of Public Health, VUB, Faculty of Medicine, Laarbeeklaan 103, 1090 Brussels, Belgium

**Keywords:** absenteeism, merger, self-certification, occupational health

## Abstract

Medical certification is often needed for absences of longer than one workday. The literature remains unclear as to whether this changes absenteeism. Earlier research found that the merging of two firms can augment or diminish short-term absenteeism. This study was conducted to examine whether prolonging self-certification or merging increases short-term absenteeism. Data from January 2014 to December 2021 were retrospectively collected from HR absenteeism files at two occupational health services in Belgium. Sickness periods of longer than 4 weeks were excluded. Company 1 started a merger in 2014, and company 2 prolonged of the self-certification period in 2018. The total full-time equivalents (FTEs) of company 1 increased by 6%, while company 2 had an increase of 28%. At company 1, there was a decline in absenteeism, while company 2 had an increase. The ARIMA (1, 0, 1) model provided a statistically significant local moving average (company 1: 0.123; company 2: 0.086) but no statistically significant parameters for the intervention (company 1: 0.007, *p* = 0.672; company 2: 0.000, *p* = 0.970). Prolonging the self-certification period by up to 5 days without medical certification or merging was not found to increase short-term absenteeism.

## 1. Introduction

Absenteeism, a worldwide problem, is defined as absence from the workplace for different reasons, for example, sickness, holiday, or parental leave. Sickness absence has a narrower definition, since the reason for the absence needs to be related to sickness [1]. The two are often used as synonyms. This study excludes leave for nonmedical reasons, such as holidays. In contrast to sickness, health is a dynamic concept that is defined by the World Health Organization as “a state of complete physical, mental and social well-being and not merely the absence of disease or infirmity”.

This study identifies short-term absenteeism as being a maximum of 4 weeks in length. This cut-off value is based on the fact that white collar employees in Belgium, a northwestern European country, receive a guaranteed salary (100%) during their first 4 weeks on sick leave, which is paid by their employer. After this period, employees receive a lower salary (60%) that is paid by the national insurance. One could argue that short-term absenteeism is financially interesting for employees. Every employee in Belgium must join the national insurance or the agency for health and disability insurance. Guaranteed income protection insurance can cover any loss of professional income.

The physician is an important ally in the preservation of health [2]. When people feel ill or unfit to work, they often consult their doctor, who can prescribe sick leave. The necessity for such medical certification (as proof of sickness by a certified professional) depends on the legal provisions of the country and/or the agreements of the enterprise. In Belgium, medical certification is necessary for absences of longer than one day. In some countries, such as the Netherlands, absence certification does not exist [3].

Some employers allow their employees to decide for themselves whether they are unable to work and need sick leave, a concept that is known as self-certification. Self-certification practices may vary not only among countries but also between workplaces within a country [4]. Generally, in Norway, a self-certification period of up to 3 days is allowed [5]. This period can be prolonged at the company level. In the public hospital sector, the self-certification period is up to 8 days [6].

The contemporary literature remains unclear as to whether prolonging the self-certification period has an effect on absenteeism [4].

Recently, the Belgian federal government made it possible for companies with more than 50 employees to change their absenteeism policy. This means that one-day absence with self-certification is permitted in order to diminish the administrative burden on medical professionals [4]. This can be applied three times a year.

Short-term absenteeism is a major problem for employers, since it not only causes a loss of productivity but also involves additional costs due to guaranteed wages and can potentially lead to the need for replacement employees in the very short term [7]. The average cost of overall sickness absence in 2010 for a European Union country was 2.5% of the gross domestic product (GDP) [8].

Sick leave is influenced by factors other than a person’s health. Factors cited as contributing to absenteeism in the international literature include gender (females > males), age above 50, overall health, job characteristics (job strain, job control), personal factors, self-evaluation of one’s functional capacity, workplace measures to support work ability, and social insurance [9,10,11,12,13].

Decisions taken at the enterprise level may also affect absenteeism. Earlier research found that the merging of two firms can have a positive or negative impact on absenteeism [14]. If the merger leads to unit-level downsizing, the short-term absenteeism will decline before the merger and rise after the downsizing process has been completed [15]. Important factors to take into account are the social support from supervisors and the possible negative impacts of the merger on job security and job type (white collar vs. blue collar). 

To improve the health and safety of employees, every employer (who employs personnel) in Belgium must join a recognized external service for prevention and protection at work. They assist the employer in carrying out various aspects of welfare legislation for which the required competencies are not internally present in the employer.

Absenteeism is a problem for everyone: employees, employers, co-workers, (national) insurance, family members, etc. This is due to factors including administrative procedures, costs, health, and general welfare.

The purpose of this study was to identify short-term variations in absenteeism as a result of major changes in which employees have no input (such as a merger or prolonged self-certification). It also aimed to establish whether self-certification could be prolonged in Belgium without an impact on short-term absenteeism and to relieve the administrative burden on the first-line medicine sector.

## 2. Materials and Methods

### 2.1. Study Population

This study included all employees from two external services for prevention and protection at work in Belgium. Company 1 has a larger employee population in Wallonia, and company 2 has a larger employee population in Flanders. In 2014, company 1 had 413.09 full-time equivalents (FTEs) and company 2 had 645.75 FTEs. At the end of the study period, the values for both companies had increased, company 1 had 441.44 FTEs and company 2 had 828.35 FTEs. At both companies, more females were employed (three times more compared to males) due to the sector being investigated as well as the favorable working hours and conditions. The total FTEs increased by 6% for company 1 and 28% for company 2. The term FTE indicates an employee’s scheduled hours divided by the hours of a full-time workweek. When an employee works 19 h per week instead of 38 h, they have worked 0.5 FTEs. 

### 2.2. Intervention Program

#### 2.2.1. Merger

Company 1 started a merger in quarter 3 of 2014. Two external services became a single legal entity that retained all employees. In the years that followed (2014–2017), the merger was implemented in all parts of the company. The absenteeism policy was standardized and tightened up, with more support from (middle) management. It was not until 2020 that the legal entity was given a definitive name to finalize the merger and to project more unity, innovation, and brand awareness.

#### 2.2.2. Self-Certification

Company 2 changed its internal HR policy in quarter 2 of 2018. The self-certification period for absences was prolonged from 1 day up to 5 days. Employees who feel sick and are unable to work have to contact their boss by telephone. Contact by telephone is the only way to officialize their absence; text-messages and and emails are not accepted.

### 2.3. Data Collection

The research data were collected retrospectively from the absenteeism files of the HR departments of two Belgian occupational health services during the period from January 2014 to December 2021. These yearly employee data files include gender, age, function, region of employment, and days of absenteeism. Pseudonymization was guaranteed by the use of employee numbers. The data were categorized into age groups (<25 years, 25–34 years, 35–44 years, 45–54 years, 55–64 years, and ≥ 65 years), function groups (medical prevention (occupational doctor and occupational nurse), nonmedical prevention (psychologist, company visitor, occupational hygienist, etc.), administration and management), and region of employment (among the 10 provinces in Belgium). 

Self-employed personnel and sickness periods of longer than 4 weeks were excluded.

Sickness absences were categorized by employee number. Absence periods were divided by week number (1–53) and number of absent working days (1–5). Partial sick days were counted as full days. National holidays were excluded from the absenteeism periods. All yearly data were assembled into one dataset in Microsoft Excel for Microsoft 365 MSO (version 2208). This dataset was transferred to IBM SPSS Statistics for Windows (version 28.0.1.1) for further statistical analysis. 

For the statistical analysis, there was a compensation for the number of sick days per FTE because of the growth in FTEs between 2014 and 2021. The following ratio was used:$$\mathrm{Sickness}\;\mathrm{Absence}\;\mathrm{Full}\;\mathrm{Time}\;\mathrm{Equivalent}\;\left(\mathrm{weekly}\right)=\frac{\mathrm{Total}\;\mathrm{sick}\;\mathrm{days}\;a\;\mathrm{week}}{\mathrm{FTE}\;\mathrm{employees}}$$

### 2.4. Statistical Methods

The Box–Jenkins autoregressive integrated moving average (ARIMA) intervention time series analysis was used to quantify the impacts of self-certification and merging on trends in the weekly rates of absenteeism. ARIMA models were used to address the inherent dynamics of the series. Coding was as follows: company 1—before merging (beforeQ3 in 2014), 0 and after merging (from Q3 in 2014), 1; company 2—certification by physician (before Q2 in 2018), 0 and self-certification (from Q2 in 2018), 1. 

To examine randomness, the Ljung–Box portmanteau test was used. 

All data analyses were conducted using IBM SPSS Statistics for Windows, version 28.0.1.1 (IBM Corp., Armonk, NY, USA); *p* < 0.05 was considered statistically significant.

Additionally, cross-correlation between the data of the two companies was determined using the cross-correlation function in SPSS. 

## 3. Results

### 3.1. Descriptive Results

Table 1 shows the descriptive statistics for this study. 

The data show a switch in age groups at company 1 from 2014 to 2021, with a large number of new employees due to the retirement of older employees. 

The HR department of company 2 could not provide 18.49% of the regions in which employees were working in 2014.

At company 1, the ratio of men to women is 1:3, while the sickness absence ratio is 1:7. At company 2, the ratio of men to women is 1:3 with a sickness absence ratio of 1:4. This implies that women are more frequently absent than men. 

As shown in Table 1, there was a rise in the percentage of sickness absence in the category ‘medical prevention’ at company 1. 

The total number of yearly sick days reduced over the years at company 1. It was assumed that merging would not have a negative effect on the total number of sick days. 

The total number of yearly sick days rose at company 2. It was assumed that self-certification would have an effect on the number of sick days, but the statistics show that this was not the case. 

With the first wave of COVID-19 (late March to early April 2020), absenteeism doubled at both companies, as can be seen in Figure 1 and Figure 2. Data from March 2020 to December 2021 were, thus, excluded from the time series analysis (described further below). 

### 3.2. Statistical Results

#### 3.2.1. Time Series Analysis

A time series model for absenteeism was developed to examine the effects of merging at company 1 and prolonging the self-certification period at company 2. 

The data used for this analysis were collected from 2014 until December 2019. Other data were excluded due to the unforeseen COVID-19 pandemic and the possibility that it could have interferes with the intervention. 

The results of the ARIMA intervention model show that the model provided a good fit for the data. The ARIMA model (1, 0, 1) had the lowest BIC (−6.473 for company 1 and −7.078 for company 2). 

The results of the Ljung–Box test showed autocorrelation coefficients that were not significantly different from zero, and the data values were found to be independent (company 1: Q = 12.337, *p* = 0.720; company 2: Q = 11.543, *p* = 0.775). 

##### Company 1

Data were collected 26 weeks before and 295 weeks after the intervention. As shown in Figure 3, there were no outliers among the observations, and variance appeared not to decrease over the time series. Therefore, no transformation was applied to the series. The 26-week preintervention series was used to identify an ARIMA (1, 0, 1) model. The local moving average parameter was 0.123 and was statistically significant (t = 7.243, *p* < 0.001). The intervention parameter was −0.007, which was not statistically significant (t = −0.424, *p* = 0.672).

##### Company 2

Data were collected for 221 weeks before and 100 weeks after the intervention. As shown in Figure 4, there were no outliers among the observations, and variance appeared not to decrease over the time series. Therefore, no transformation was applied to the series. The 221-week pre-intervention series was used to identify an ARIMA (1, 0, 1) model. The local moving average parameter was 0.086, and this was statistically significant (t = 15.638, *p* <0.001). The intervention parameter was 0.000 and was not statistically significant (t = −0.038, *p* = 0.970).

#### 3.2.2. Cross-Correlation

Cross-correlation was used in this study to compare the similarities in the absenteeism data of the two studied companies. The total number of sickness days per company cross-correlated with the FTE/week gave a correlation of almost 1 (company 1 = 0.756 and company 2 = 0.982, *p* < 0.001). This was not surprising, given that the dates roughly matched. The cross-correlation between the two companies could not be used for further analysis, since the value was far from 1 and was not statistically significant (0.428, with *p* > 0.05), as seen in Table 2. 

## 4. Discussion

Absenteeism is a hot topic, since it has a substantial financial impact on corporations [3]. Employee expenses are often reported as being one of the main business expenses. At company 1, absenteeism was reduced from 6.9% (2014) to 5.11% (2021). This was due to changes in the company culture and absenteeism policy. At company 2, an increase from 3.85% per FTE to 5.28% was observed. This may have been related to the internal circumstances of the company, such as higher work stress, the working atmosphere, or a change in culture.

The data in this study confirm the finding of earlier research [14] that merging with unit-level upscaling reduces absenteeism. The merger was not accompanied by job insecurity; on the contrary, it led to more favorable working conditions, unity, and innovation. Another reason as to why absenteeism did not increase could be that employees experienced good support from their supervisors.

Company 2 decided to prolong the self-certification period for absences to up to 5 days in Q2 of 2018. Employees decide whether they are able to work or not. Arguably, the telephone contact with superiors could lead to less absenteeism, since the barrier to absence is higher due to the personal nature of such contact. They need to explain to their boss the reason as to why they are not attending work. This mechanism may be more discouraging. Another effect of this method could be more presenteeism, since more dedicated employees might come to work when sick. It is tougher to call in sick when one has a ‘bad day’ or does not feel like working on a Monday morning. On the other hand, this might result in reduced work productivity [16] and lead to more frequent absenteeism in the future [17]. Bad health and low job control present a low barrier for absence. These parameters should be ameliorated to reduce absenteeism. Presenteeism could lead to additional costs for employers in the end; thus, it is not necessarily positive for the employer or the employee [18]. 

Another important factor to take into consideration is the degree of education [19]. In this study, only white-collar employees, the majority of whom had a higher education degree, were included. However, blue-collar workers do a higher amount of heavy manual work [20], which could influence absenteeism negatively. This could have led to a selection bias.

This study does not specify the reasons for short absences. People with poorer health should not be victimized by this method, so a solid absenteeism policy is necessary. If certain individuals are absent more frequently than others, an individual approach is essential. These people could benefit from more support from their supervisor or more help from a colleague to look for a solution (e.g., working part time or fewer hours) or other types of social leave. It is possible that certain individuals are absent from the workplace due to the sickness of family members, especially in the case of those with small children who cannot stay home alone, as it is not always easy to find another solution in a short period of time [21].

Partial sick days were counted as full days. This could give a slightly distorted view, since more females are employed at both companies. This can be clearly seen in Table 1. In general, females tend to work part time more often, which could lead to an overestimation of the number of sick days taken by females [10,12,13].

Data from March 2020 to the end of 2021 were excluded, since this was during the COVID-19 pandemic. This was a highly unusual situation, with more frequent absences at the start of the pandemic because of the knowledge gap regarding the disease. Employers, governments, and doctors preferred to err on the side of safety before the testing capacity could be adapted to the needs in Belgium. Many employees with mild symptoms stayed home, whether their symptoms were related to COVID-19 or not.

Working from home may have had a positive influence on work productivity. Working from home means not having to get up early in the morning to go to work (and thus, allowing more sleep and recovery time), skipping traffic jams [22], and spreading the workload throughout the workday and even working outside of regular working hours. This higher level of job control can lead to more productivity and less absenteeism. In the study population, some employees were able to work from home and had mild symptoms or did not feel 100% fit. This also could have interfered with the data.

The pattern of absenteeism shown in this study differs from that shown for other European countries [23]. Studies in other EU countries show two recurring annual peaks between 2015 and 2021, around mid-August and at the end of December. This typical increase in absenteeism is less visible in the data.

The data show no peak during the summer, contrary to the previously established EU data [16], arguably due to the summer holiday period. Data for company 1 showed a decline in absenteeism at the end of the year, while data for company 2 showed a limited increase. During the winter in Belgium, there is an increase in flu and other cold viruses.

In the 2020 EU data, there is a large peak at the end of March (weeks 12–13). This peak is four times higher compared with the previous four weeks and can be entirely attributed to the COVID-19 pandemic and a corresponding increase in absenteeism at the workplace. Both companies showed the same increase in data (Figure 1 and Figure 2).

This study may be limited due to the use of retrospective data. No additional information could be collected. Another drawback of this study is the unforeseen impact of the global COVID-19 pandemic.

The strengths of the study lie in the large amount of data collection over an 8-year time period and the inclusion of all employees from two of the largest occupational health services in Belgium.

## 5. Conclusions

This study demonstrates that prolonging the self-certification period has no impact on absenteeism. Furthermore, this study concludes that the merging of two firms has no impact on absenteeism. Crucial factors are the more favorable working conditions that were achieved by the merger presented in this article. Merging may lead to a different impact on absenteeism when it leads to reduced terms of employment or layoffs. This is an interesting area of future research.

Prolongation of the self-certification period can be implemented in Belgium without an augmentation of short-term absenteeism in companies with the same working conditions. More prospective studies are needed in the future to ensure findings can be also implemented in sectors with different terms of employment.

## Figures and Tables

**Figure 1 ijerph-20-03660-f001:**
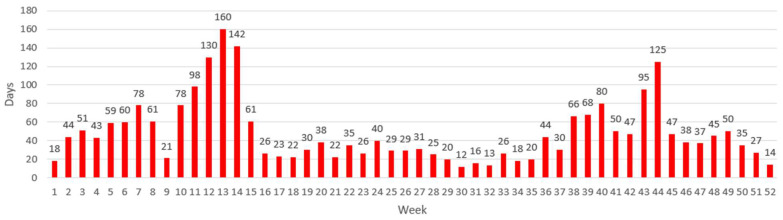
Weekly absences for company 1 in 2020, showing an increase from week 11, which was the beginning of the COVID-19 pandemic in Belgium. This increase continued for 4 weeks during the first COVID wave. A second peak started from week 43, which could have been partly due to another wave of infections at the workplace.

**Figure 2 ijerph-20-03660-f002:**
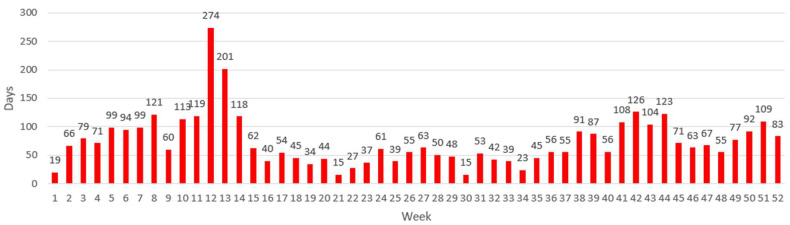
Weekly absences for company 2 in 2020, Showing an increase starting from week 12, which was the beginning of the COVID-19 pandemic in Belgium, and this rise continued, marking the first COVID wave. The rest of the year was more stable, although a small rise can be observed starting from week 41, marking the second wave of COVID infections in Belgium.

**Figure 3 ijerph-20-03660-f003:**
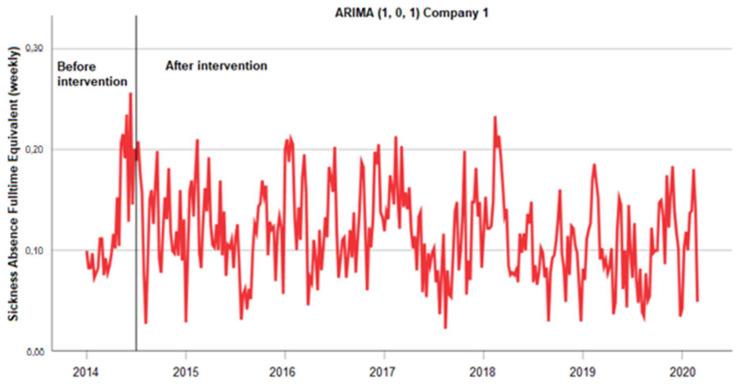
Absenteeism data were collected at company 1 for 321 weeks. After 26 weeks, two firms started a merger (=intervention). The first 26 weeks were used as a baseline period to identify an ARIMA (1, 0, 1) model. The local moving average parameter was 0.123, which was statistically significant (t = 7.243, *p* < 0.001). The intervention parameter −0.007 was not statistically significant (t = −0.424, *p* = 0.672).

**Figure 4 ijerph-20-03660-f004:**
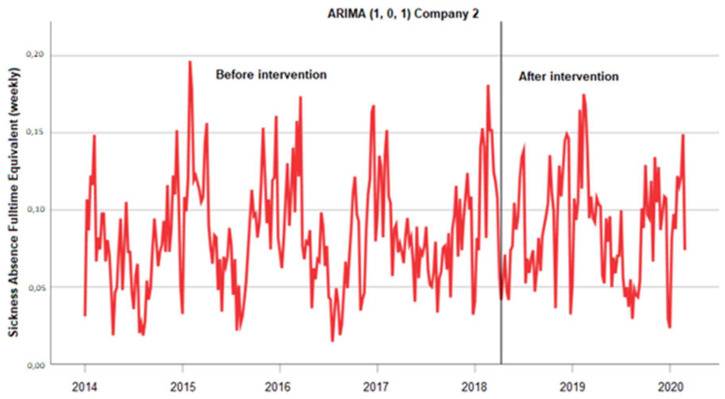
Absenteeism data were collected at company 2 for 321 weeks. After 221 weeks, the absenteeism policy was modified. Self-certification was prolonged from 1 day to 5 working days (=intervention). The 221-week pre-intervention was used as a baseline period to identify an ARIMA (1, 0, 1) model. The local moving average parameter was 0.086, and it was statistically significant (t = 15.638, *p* < 0.001). The intervention parameter 0.000 was not statistically significant (t = −0.038, *p* = 0.970).

**Table 1 ijerph-20-03660-t001:** Characteristics of absenteeism data from two occupational health services in Belgium in 2014, 2019, and 2021. The table shows the absolute numbers and percentages of absentees in the two companies categorized by gender, age, function, and region.

KERRYPNX	Company 1						Company 2					
	2014 n = 1147		2019 n = 923		2021 n = 835		2014 n = 1087		2019 n = 1626		2021 n = 1811	
Total sick days (number of days)	2850		2374		2256		2483		3576		4377	
FTE	413.09		441.47		441.47		645.75		781.9		828.35	
Total sick days per FTE (%)	6.90		5.38		5.11		3.85		4.57		5.28	
	n	%	n	%	n	%	n	%	n	%	n	%
Gender of absentees												
Men	153	13.34	147	15.93	127	15.21	272	25.02	413	25.40	477	26.34
Women	994	86.66	776	84.07	708	84.79	815	74.98	1213	74.60	1334	73.66
Age of absentees (years)												
<25	20	1.74	10	1.08	20	2.40	39	3.59	18	1.11	46	2.54
25–34	186	16.22	240	26.00	218	26.11	303	27.87	517	31.80	474	26.17
35–44	465	40.54	222	24.05	226	27.07	395	36.33	512	31.49	594	32.80
45–54	323	28.16	266	28.82	197	23.59	220	20.24	368	22.63	462	25.51
55–64	148	12.90	182	19.72	165	19.76	124	11.41	194	11.93	233	12.87
≥65	5	0.44	3	0.33	9	1.08	6	0.55	17	1.05	2	0.11
Function of absentees												
Administration	532	46.38	442	47.89	279	33.41	276	25.39	412	25.33	415	22.92
Medical prevention	387	33.74	320	34.67	356	42.63	539	49.59	778	47.85	925	51.08
Non-medical prevention	177	15.43	109	11.81	140	16.77	272	25.02	436	26.81	470	25.95
Management	51	4.45	45	4.88	49	5.87	0	0	0	0	1	0.06
Unknown	0	0	7	0.76	11	1.32	0	0	0	0	0	0
Region of absentees												
Antwerp	25	2.18	39	4.23	28	3.35	280	25.76	575	35.36	572	31.58
Brussels	367	32.00	296	32.07	326	39.04	70	6.44	191	11.75	239	13.20
Limburg	9	0.78	9	0.98	16	1.92	101	9.29	126	7.75	194	10.71
East Flanders	33	2.88	65	7.04	58	6.95	131	12.05	193	11.87	221	12.20
West Flanders	7	0.61	13	1.41	12	1.44	77	7.08	161	9.9	145	8.01
Flemish Brabant	14	1.22	13	1.41	1	0.12	196	18.03	320	19.68	334	18.44
Luxembourg	31	2.7	1	0.11	10	1.20	0	0	0	0	0	0
Liège	425	37.05	217	23.51	192	22.99	0	0	0	0	0	0
Hainaut	184	16.04	228	24.70	150	17.97	0	0	0	0	0	0
Namur	52	4.53	42	4.55	41	4.91	31	2.85	57	3.51	99	5.47
Walloon Brabant	0	0	0	0	0	0	0	0	0	0	0	0
Unknown	0	0	0	0	1	0.12	201	18.49	3	0.18	7	0.39

n, number of absentees; FTE, full time equivalent.

**Table 2 ijerph-20-03660-t002:** Estimated cross-correlation between the total number of sickness days per company and the FTE/week.

Lag	Cross	Std. Error *
−7	−0.035	0.056
−6	−0.017	0.056
−5	−0.056	0.056
−4	−0.016	0.056
−3	−0.090	0.056
−2	−0.231	0.056
−1	−0.348	0.056
0	−0.428	0.056
1	−0.261	0.056
2	−0.203	0.056
3	−0.032	0.056
4	−0.046	0.056
5	−0.034	0.056
6	−0.039	0.056
7	−0.027	0.056

* (Based on the assumption that the series are not cross-correlated and that one element of the series is white noise).

## Data Availability

Not applicable.

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
