# Peer review of "Absenteeism at Two Occupational Health Services in Belgium from 2014 to 2021"

_ijerph, 2023, doi:10.3390/ijerph20043660_

Round 1

Reviewer 1 Report

Thank you for the opportunity to read a very interesting text. The topic might be of interest, but unfortunately, the manuscript under review is below acceptable quality standards in terms of conclusions. The last paragraph about conclusions and directions for further research should be expanded because there are some issues raised that could be pursued in further work.

This section of Conclusions is too synthetic and does not refer to the practical and theoretical applicability of the study, as well as to the future directions by which it can be continued.

Kind regards!

Reviewer 2 Report

-Introduction is poorly organized. It presents few random informations but it does not provide an international contest and an updated state of the art regarding absenteism. The aims of the study is missing, therefore when proceeding in the reading the reader has not any clue regarding the real subject of the paper. I strongly recommend a complete re-writing of the Introduction. 

-Methods. Also methods are not clear at all, and it is mainly due to the fact that there is no description of study's aims and scope. 

-Define FTE

-What job sector did the selected companies belong? It is not clear. 

-3.1.2 what do you mean with "factors"? it is not clear the purpose of the paragraph 

-3.1.3 is not useful for the presentation of results 

-Discussion provides some points of views of the authors, but it lack of comparisons with preovious studies and justifications for many hypothesis (such as telephonic contact). 

-par. 2.2.2 needs to be more specific, they are difficult to understand.

Reviewer 3 Report

The article entitled: "Absenteeism in two occupational health services in Belgium from 2014 till 2021" is interesting and has great potential. Authors need to make some changes to help improve the submitted manuscript. 

-        Abstract: need to be edited. The author’s state: "Earlier research found that merging of two firms can have positive or negative impact." What impact? Impact on what? I recommend changing the wording of the main aim of the article (has any effect – bad wording for a scientific article). The main aim of the article must be the same throughout the manuscript!

-        Introduction: scientific sources need to be supplemented. In its current form, the introduction does not provide a sufficient theoretical basis! Use the scientific databases WOS and SCOPUS, explain the reasons for the study. It is not enough to state: (There were only a few studies carried out on the topic and that's the main reason for this study.) = what studies? Who is their author? Where did they originate? Where were they published?

-        Table 1: Characteristics of absenteeism data from two occupational health services (page 4 and 5, I recommend the authors to add a brief description of the Table 1).

-        The discussion generally serves to confront (discuss) the obtained results with other works (studies). In the submitted manuscript, only three sources (15, 16 and 17) are listed in the discussion section. This fact has a negative impact on the quality of the discussion.

-        5. Conclusions – I consider line 335-338 insufficient! It is necessary to supplement the conclusions. I recommend moving the section on future research from the discussion section to the conclusions section. Furthermore, it is necessary to add a part of the limitation of the research. At the same time, I recommend the authors to add recommendations for practice.

Final evaluation: In my opinion, the submitted manuscript is interesting, but requires additions and modifications. From a formal point of view, I encourage authors to double-check the IJERPH Author Guidelines (ISSN 1660-4601). I wish the authors the best of luck.

Round 2

Reviewer 2 Report

Dear authors,

thank you very much for having addressed my comments. I think now the paper has really improved. only I suggest in line 89: It also establishes if prolonging self-certifi- 89 cation can be implemented in Belgium...I believe it is too generic as an aim for a paper, please be more specific.

Best regards. 

Author Response

Thanks for reviewing. See attachement.
